# The Effect of Fish Oil-Based Foods on Lipid and Oxidative Status Parameters in Police Dogs

**DOI:** 10.3390/biom12081092

**Published:** 2022-08-08

**Authors:** Branko Ravić, Jasmina Debeljak-Martacić, Biljana Pokimica, Nevena Vidović, Slavica Ranković, Marija Glibetić, Predrag Stepanović, Tamara Popović

**Affiliations:** 1Center for Excellence in Food and Metabolism, Institute for Medical Research, University of Belgrade, 11000 Belgrade, Serbia; 2Faculty of Veterinary Medicine, University of Belgrade, 11000 Belgrade, Serbia

**Keywords:** bioactive lipids, fatty acids, omega-3, oxidative stress, working dogs

## Abstract

The synthesis, degradation, and reconstruction of the cell membrane as a metabolic pathway of phospholipids is a constant and dynamic process. Fatty acids as bioactive lipid components of plasma and erythrocyte phospholipids as structural lipids have biological roles in the integrity of cell membranes. Fatty acids, depending on the chain length, the degree of saturation, and the synthesis pathways, can alleviate inflammation and oxidative stress caused by excessive exercise. Considering that changing food intake or diet can influence fatty acid phospholipid metabolism, our study aimed to determine the potential benefits of fish-based diets in working (police) dogs undergoing intensive training concerning bioactive lipids such as fatty acids, phospholipids of plasma, and erythrocytes. Fatty acid esters’ composition of plasma and erythrocyte phospholipids as a bioactive lipids, in addition to markers of oxidative stress and metabolic parameters, were analysed by GC chromatography. The food was well tolerated by all dogs, and the compliance to the diet was high throughout the study. After the treatment with fish-based food, blood glucose, total, and LDL cholesterol levels were significantly reduced, indicating positive biochemical profiles of dogs. Correlations of fatty acid phospholipid compositions between plasma and erythrocytes have shown that both plasma and erythrocytes could represent markers of omega-3 eicosapentaenoic and docosahexaenoic acid intake levels in dogs. Morover, fish-based food supplementation caused a significant reduction in lipid peroxidation markers. The enrichment of dogs’ diets with marine fish could improve oxidative status and improve roles and status of bioactive lipids, such as membrane phospholipids and fatty acids, as its components in plasma and erythrocytes in police dogs exposed to intensive exercise.

## 1. Introduction

Police dogs are working dogs that are exposed to intense running exercises. These activities could cause intensive stress to the animals. One of the approaches for preventing stress in animals is to enrich the dogs’ diet with ingredients that can mitigate the negative health effects of intensive training.

Some authors suggest that fatty acids might influence inflammation and oxidative stress, depending on the chain length and the degree of saturation, although there is little information regarding the optimal fat intake for canine athletes [1]. Certain results indicate the beneficial effect of eicosapentaenoic acid (EPA), docosahexaenoic acid (DHA), and omega-3 long-chained polyunsaturated fatty acids (LC PUFA) on canine health [2]. Studies have shown that the supplementation of food-derived alpha-linolenic acid (ALA), the dietary precursor of EPA and DHA, might lead to the accumulation of EPA and docosapentaenoic acid (DPA), but rarely of DHA in dogs [3,4]. Therefore, effective supplementation with omega-3 LC PUFA could only be achieved by enriching dogs’ diets with fish [5]. In canines, the beneficial effects of omega-3 LC PUFA supplementation have been found to reduce heart disease, chronic renal failure, cancer, atopic dermatitis [6] as well as degenerative joint diseases [7]. Data concerning the effects of fish-based food on lipid profile in dogs are scarce. On the other hand, a large number of human studies’ data have shown a positive impact of a fish-based diet [8].

Even though exercise shows many health benefits, on the contrary, intense and rigorous exercise can result in the excessive generation of free radicals [9], resulting in oxidative damage to proteins and lipids in the contracting myocytes [10]. Intensified periods of physical training with minimal recovery may perturb redox homeostasis, inducing a state of chronic oxidative stress and inflammation [11]. As a result, chronic oxidative stress could limit physiological adaptations to exercise. The antioxidant mechanisms of trained dogs may, in some instances, be inadequate to meet the antioxidant requirements of repetitive endurance exercises [12]. Research indicates that the antioxidant supplementation of sled dogs may attenuate exercise-induced oxidative damage [13].

Dietary fish oil is efficient in reducing the in vivo oxidative damage of proteins [14]. According to some researchers, DHA and EPA can inhibit cell death via the induction of different antioxidants and thus protect cells [15]. Luostarinen and coworkers [16] showed that dietary omega-3 fatty acids could increase superoxide dismutase activity.

In addition, omega-3 PUFA could probably improve athletic performances by modulating membrane permeability and insulin sensitivity, which could make muscle cells more permeable for valuable nutrients, such as glucose and amino acids [9]. This is supported by the upregulation of the GLUT4 transporters, so the regular consumption of omega-3 FA could result in a metabolic stimulator of muscle cells. Omega-3 significantly induced metabolic genes as well as oxidative metabolism (oxygen consumption), glycolytic capacity (extracellular acidification), and metabolic rate [17].

This study aims to evaluate the effects of supplementation with fish-based food on the lipid and oxidative status of police dogs subjected to intensive training.

## 2. Materials and Methods

### 2.1. Study Design

The research was carried out in accordance with all relevant national regulations and institutional policies for the care and use of animals (Ministry of Education, Science and Technological Development number 451-03-68/2022-14/200015 from date 2 April 2022. and filed in Ethical Committee of the Faculty of Veterinary Medicine by 17 June 2017, number 01-592. The ARRIVE guidelines for reporting in vivo animal research were followed. 

A signed consent form was received from the police to use their working dogs in our experiment, but this agreement represents a document that is not available to some other parties.

Fifteen working dogs of the Belgian Shepherd (Malinoa) breed were included in the study, which was the maximum obtainable number of dogs, so for this reason no FFcontrol group was available. Dogs aged from 3 to 7 years. Water was provided *ad libitum*. The individual daily food intake of dogs was determined by measuring bowls before and after feeding. The period of study was in summer when the average daily temperature was not above 35 degrees.

Based on the results of the physical examination (blood pressure, pulse, temperature, breathing, skin condition, and skin cover) the dogs were in an almost satisfactory health condition. According to the constitution, dogs belonged to category number 3 (condition state), meaning that they had a weight that corresponded to this breed of dogs. The basic criterion for not including dogs in the study was the presence of any form of inflammation. All dogs were trained three times a day for one hour. During the research, the dogs had a defined program of intensive exercise (walking, trotting, and galloping) with predicted jumps and jumping hurdles. They were localized in police kennels. The dogs were fed a commercial basic maintenance diet (Farmina dog foods) that meets the needs of dogs in basic nutrients (vitamins) for a month. This pre-trial food was almost completely deficient in omega-3 LC PUFA, EPA, and DHA. After that animals were assigned to a commercial fish oil-based food especially enriched with fish oil (Farmina dog foods) for three months. The foods had almost the same fat content, but the origin of the fat was quite different. Diet 1 was enriched with animal fats, while Diet 2 had only fish oil as a source of animal fat. Food composition data, as well as the FA profile of both diets, are presented in Table 1. Both types of diets were supplemented with vitamins.

### 2.2. Blood Collecting and Lipid Analyses

Blood samples (3 mL or less) were collected from *vena cephalica antebrachii* into EDTA-containing vacutainer tubes. Blood samples were taken 12 h after fasting. Plasma and erythrocytes were isolated from full blood by standard procedure. The aliquots were kept frozen at −80 °C until further analysis. Routine biochemical analyses were performed in plasma samples using a Cobas c-111 biochemical analyzer (Roche, Basel, Switzerland). The blood was taken three times during experiments: at the beginning, in the middle, and at the end of the experiment. The results of the second and third measurements did not differ significantly, so only the results of the last measurement were shown.

The total lipid extracts from plasma were isolated by the method described by Folch et al. [18] with some modifications. Namely, total lipid extracts from plasma were prepared by adding chloroform/methanol (2:1, *v*/*v*) mixture. The total lipid extracts from erythrocytes were isolated by the mixture of chloroform/isopropanol (7:11, *v*/*v*) according to the method described by Rose and Oklander [19].

Phospholipid separation from other lipid subclasses in lipid extracts was done on a silica thin-layer chromatography plate using the solvent system of petroleum ether, diethyl ether, and glacial acetic acid (87:12:1, *v*/*v*/*v*). Fatty acid methyl esters were prepared by transmethylation with 3N HCl in methanol. Fatty acid methyl esters were analyzed by gas-liquid chromatography in a Shimadzu chromatograph GC 2014 (Kyoto, Japan) equipped with an injector and a flame ionization detector on Rtx 2330 column (60 m × 0.25 mmID, film thickness of 0.2 μm, RESTEK, Bellefonte, PA, USA). The identification of FA methyl esters was made by comparing sample peak retention times with standards (PUFA-2, Supelco, Bellefonte, PA, USA). Finally, individual FA was expressed as a percentage of the total identified FA.

Total carbohydrates (TCH) content, crude “by difference”, was calculated according to following formula: TCH (%) = 100% − % (Crude Protein + Ash + Crude Fat + Moisture). Analytical analysis of proximates in the diets was done in referent laboratory using referent methods and presented in percentage. The ash content was measured by the direct gravimetric method (AOAC 923.03) [20] representing the ashing of the samples in an oven at 550 °C until a constant weight was attained. The crude protein (CP) content was determined by Kjeldahl method [20] published in 1883, which has traditionally been the accepted reference method for the determination of protein in dairy products (AOAC 955.04D) [19]. Total fat in food was determined by acid hydrolysis described in AOAC Official Method 922.06 [20]. Total minerals were determined by atomic absorption spectroscopy (AAS) (Varian spectra AA-10, Agilend tehnologies, Santa Clara, CA, USA).

### 2.3. Determination of Superoxide Dismutase Activity

The superoxide dismutase (SOD) activity was determined in erythrocytes using the Ransod kit (Randox, Crumlin, UK). The reaction was based on the superoxide radical anion production by the action of xanthine oxidase which reacted with 2-(4-iodophenyl)-3-(4-nitrophenol)-phenyltetrazolium chloride, resulting in the formation of a red formazan dye. The higher SOD activity in the sample, the less formazan dye produced. One unit of SOD was defined as the amount of enzyme resulting in 50% inhibition of dye formation [21].

### 2.4. Determination of Glutathione Peroxidase Activity

The glutathione peroxidase (GPx) activity was measured in erythrocytes using the Ransel kit (Randox, Crumlin, UK). Reduced glutathione (GSH) was oxidized in the presence of cumene hydroperoxide resulting in the formation of glutathione-disulfide (GSSG). Glutathione reductase then reduced the GSSG to produce GSH in the presence of NADPH coenzyme. One unit of GPx was defined as nanomoles of NADPH oxidized per minute and calculated based on the NADPH molar absorption coefficient [22].

### 2.5. Determination of Catalase Activity

The catalase (CAT) activity was determined in erythrocytes by the partially modified method described by Aebi [23]. The reaction consisted of degradation of hydrogen peroxide—H_2_O_2_ (Sigma Aldrich^®^, Darmstadt, Germany) by endogenous catalase. It was detected as the decrease in absorbance at 240 nm. One unit of CAT activity was defined as the amount of enzyme which decomposes 1 μmol of H_2_O_2_ per minute. The activities of catalase, GPx, and SOD enzymes were expressed in U/gHb, and cyanmethemoglobin method with Drubkin’s reagent was applied for the assessment of hemoglobin (Hb) concentration.

### 2.6. Determination of Lipid Peroxidation

The lipid peroxidation was measured as the levels of thiobarbituric acid–reactive substances (TBARS) which were determined in plasma samples obtained at the baseline and at the end of the three-month period of feeding dogs with fish oil-enriched food. The analysis was performed according to the previously published method [24]. The calibration curve was generated with malonaldehyde bis (dimethyl acetal) using standard. The levels of TBARS were expressed in MDA equivalents.

### 2.7. Statistical Analyses

The variables were tested for normality by Shapiro–Wilk test. For normally distributed variables, a repeated-measures *t*-test was used and data were expressed as mean ± standard deviation. When the variables were not normally distributed, Wilcoxon signed-rank test was applied and data were presented as median (min-max). A 2-tailed *p*-value <0.05 was considered statistically significant. (Graphpad prism, version 4). Spearman’s correlation coefficient is a statistical measure of the strength of a monotonic relationship between paired data.

## 3. Results

### 3.1. Biochemical Analysis

The change in dietary food intake was well tolerated by all dogs. Compliance with the diet was high throughout the study. The body weight of the animals did not change significantly during the study. On average, the dogs consumed the same amounts, between 300–350 g per day of food during the entire study period. Importantly, the dogs maintained their body weight during the study (28 ± 0.75 vs. 29 ± 0.8 kg).

After the treatment with fish-based food, as a consequence of fish oil consumption, the levels of blood glucose, total, and LDL cholesterol were significantly reduced (Table 2).

Triglyceride values remained unchanged.

### 3.2. Fatty Acid Composition

The quantity of saturated fatty acids (SFA) significantly increased in plasma, but the change was insignificant in erythrocyte membranes. Monounsaturated fatty acids (MUFA) were significantly increased in both plasma (Table 3) and erythrocytes (Table 4), indicating that omega-3 fish-based diet might increase MUFA, and consequently lipogenesis. The increase in omega-3 LC PUFA during the feeding period was compensated by a decrease in omega-6 LC PUFA. The total PUFA also decreased. While EPA and DHA increased significantly, arachidonic acid (AA) and linoleic acid (LA) notably decreased. Accordingly, the omega-6/omega-3 ratio was significantly lower after the treatment. The results of the fatty acid profile in the phospholipids of plasma are given in Table 3, while the results of the FA profile of erythrocyte phospholipids are shown in Table 4.

The correlations of omega-3 phospholipid fatty acid composition between plasma and erythrocytes showed that both plasma and erythrocytes could represent markers of omega-3 intake from food in dogs during the previous three-month period (Table 5).

### 3.3. Oxidative Stress Parameters

Significantly higher activities of GPx and CAT were seen after the treatment. The level of lipid peroxidation drastically decreased after supplementation as a possible consequence of higher activities of these enzymes. These results are given in Table 6.

## 4. Discussion

In the present study, we compared the incorporation of omega-3 LC PUFA into plasma and erythrocyte phospholipids in dogs fed a commercial diet containing fish and fish oil. Initially, the commercial diet that the dogs consumed before the experiments was almost without omega-3 LC PUFA. Also, the study was designed to explore if fish-based foods can alleviate oxidative stress induced by the intensive training of police dogs. Biochemical analysis showed a significant decrease in glucose, total, and LDL cholesterol levels. Most animal experiments showed positive effects of omega-3 LC PUFA on insulin sensitivity and glucose metabolism [25]. Using animal models, it has been shown that fish oil or individual omega-3 PUFA were implicated in the positive regulation of insulin resistance [26]. Dogs are prone to hyperlipidemia, as it has been recently recognized as an important problem in the dog population [27]. It includes complications such as pancreatitis, liver disease, and atherosclerosis. The metabolic parameters of dogs significantly improved which was especially important for those with increased plasma cholesterol concentrations [28]. The literature data on the effect of omega-3 LC PUFA on lipid status in police dogs are limited.

Recently, it has been shown that dogs are prone to hyperlipidemia. It was interesting that fish intake (adequate content of FO is achieved by adding both whole fish and fish oil to the food, respectively) had a more pronounced effect. Our fish oil-based food gave us the same result, considering LDL decrease. In line with our results, the effect of omega-3 LC PUFA on triglycerides is rarely seen in rabbits, monkeys, dogs, and humans [29]. Monounsaturated fatty acids (MUFA) and omega-3 LC PUFA significantly increased in both plasma and erythrocytes. The increase in omega-3 LC PUFA during the feeding period was compensated by a significant decrease in omega-6 LC PUFA. The total PUFA was also significantly reduced in both plasma and erythrocytes. While EPA and DHA significantly increased, arachidonic (AA) and linoleic acid (LA) were significantly reduced, notably lowering the omega-6/omega-3 ratio after the treatment, which supported the fact that omega-3 LC PUFA represented major agent-inducing changes. This is a well-known competitive enzyme mechanism for the same substrates between omega-3 and omega-6 families of FA.

Similar findings are reported by others; diets containing different amounts of omega-3 LC PUFA led to increased values of these PUFA in the erythrocyte membrane (EM) [30]. The incorporation of omega-3 PUFA in dogs fed fish-enriched commercial diets gradually increased, reaching a plateau within eight weeks [30,31]. The rise in omega-3 FA (mainly EPA and DPA) was compensated by a decrease in omega-6 PUFA, especially LA and AA. The high EPA content in the fish oil-enriched diet possibly reduced AA incorporation in plasma and erythrocyte phospholipids [6]. Dogs fed high fish-based food had higher plasma EPA and DHA levels and lower concentrations of plasma total omega-6 FA, LA, and AA than dogs fed low omega-3 PUFA diets [32]. Jude [33] showed that a fish oil-enriched diet resulted in the preferential incorporation of EPA and, to a lesser extent, of DHA, at the expense of AA, in the plasma phospholipids, erythrocyte phospholipids, and cardiomyocyte phospholipid fractions of dogs [34]. EPA and DHA are inversely correlated with inflammation [34], a process following intensive training. EPA and DHA give rise to anti-inflammatory and inflammation-resolving mediators. Low omega-3 index (sum of EPA and DHA) is also a risk factor for the development of CVDs [35]. The cardiovascular benefits of omega-3 PUFA could originate from their ability to improve lipid metabolism and reduce the synthesis of pro-inflammatory eicosanoids derived from omega-6 PUFA. At an adequate level of incorporation, EPA and DHA influence the membrane fluidity as well as membrane protein-mediated reactions, the generation of lipid-mediators, cell signaling, and gene expression in different cells [36]. Dogs have the capacity to metabolize omega-3 fatty acids, and there are visible effects of these fatty acids on their skin and coats, inflammatory responses, as well as the neurologic development of puppies [37]. During our study, the coats of dogs significantly improved, which was confirmed to a high degree by a group of professors of veterinary medicine.

In our study, the decrease in omega-6/omega-3 ratio was highly significant, both in plasma and erythrocytes phospholipids. The ratio in erythrocytes changed from 29.81 to 3.24. In their study, Stoeckel and coworkers underlined that changes in omega-6 and omega-3 PUFA in erythrocyte phospholipids modified the omega-6/omega-3-ratio [32]. Numerous epidemiological and clinical studies have shown that an imbalance in the omega-6/omega-3 ratio promotes the pathogenesis of many diseases [37]. The balance between omega-6 and omega-3 PUFA is an important factor involved in the prevention of CVD and atherosclerosis through a possible improvement of stress parameters [37]. By correlating some plasma (Table 3) and erythrocyte (Table 4) phospholipids’ FA composition before and after the treatment, we investigated whether FA from both plasma and erythrocytes could represent markers of short-term dietary omega-3 intake. Our results clearly showed that changes in the content of EPA, DHA, and total omega-3 in phospholipids in plasma and erythrocytes after three months of treatment were significantly positively correlated, meaning that changes in omega-3 LC PUFA in both plasma and erythrocytes could be representative biomarkers of omega-3 intake, as confirmed by Stoeckel and coworkers [32]. We indicated that a high omega-3 index could be the main marker of fish intake in dogs, but maybe in general species. It is important to underline that the omega-3 index as well as the ratio of AA/EPA significantly improved after feeding with fish-based food, indicating that fish-based food significantly reduced inflammation in general. EPA is a key anti-inflammatory LC-PUFA. Conversely, omega-6 FA arachidonic acid is a precursor to a number of proinflammatory mediators. EPA acts competitively with AA for the key cycle oxygenase and lipoxygenase enzymes to form less inflammatory products [37,38]. The EPA/AA ratio has clinical utility in cardiovascular disease [39] while the AA/EPA ratio can be used to better evaluate inflammation and lipid content of cell membranes [40].

Physical exercise represents an important factor involved in the production of free radicals and reactive oxygen species due to the increase in oxygen consumption in the body [40]. Repetitive endurance exercises in dogs have been shown to increase lipid peroxidation [41]. The study of Venkatraman [42] suggests that feeding FO increases the activity of liver cytosolic catalase and GSH-Px rats. In line with our results, exercise significantly decreased the generation of TBARS in the liver microsomal lipids.

Fish oil-based foods enabled the efficient incorporation of omega-3 LC PUFA in plasma and erythrocyte phospholipids which resulted in a significant reduction in the generation of lipid peroxidation products. Our preliminary but very relevant results showed that, after the enrichment of dogs’ food with fish, markers of lipid peroxidation decreased, while enzymes included in the degradation of superoxide anion radical, hydrogen peroxides, and lipid hydroperoxides significantly increased. One explanation could be that omega-3 LC PUFA is sensitive to oxidation, which could be the reason for increased enzyme activities involved in the degradation of lipid peroxides and decreased lipid peroxidation. On the other hand, it is important to emphasize that after incorporation into the cell membrane, EPA and DHA occupy a specific conformation that is more stable and less susceptible to oxidative stress. However, considering that these dogs were under severe stress, TBARS production could be increased and, in that way, induce a rise of enzyme activities in circulation in order to decrease oxidative damage, like reducing the aforementioned TBARS.

The TBARS decrease is a very important result. We offered two contradictory explanations, but we believe that is not complete, and that is the reason why we consider an unexpected reduction in TBARS to be a limitation of the study. Some other authors testify that omega-3 LC PUFA has a direct antioxidative activity [43]. PPARα activation is a mechanism by which fish oil omega-3 PUFA enhances mitochondrial fatty acid oxidation and antioxidant capacity in humans [44]. Fish oil treatment significantly improved plasma lipid profile, liver FAs composition, and the parameters of oxidative stress in male Wistar rats [38]. Of all the cellular components compromised by the harmful effects of ROS, the cell membrane is the most severely affected by lipid peroxidation, which leads to changes in the membrane structure and permeability. After feeding with fish oil-based food, we observed significantly higher activities of GPx and CAT followed by a decrease in lipid peroxidation. Repetitive exhausting training causes a depletion in plasma antioxidant concentrations [12]. Fish-based diets have been shown to enhance the GPx activity of platelets and erythrocytes in both animals and humans. So, increased enzyme activity could be a physiological mechanism that weakens the generation of free radicals under oxidative stress in animals exposed to intensive exercise with minimal recovery [44].

Omega-3 LC PUFAs as components of fish-based foods could bring various benefits to athletes by attenuating the generation of stress and inflammation induced by excessive training, thus being a promoter of improved muscle performance and immune function [9]. However, if dogs are exposed to intensive physical exercise, markers of oxidative stress significantly are raised due to the high level of ROS that probably cannot be completely neutralized by their own antioxidant defenses [44]. Fish-based diets, especially components like omega-3 LC PUFA, in a still unknown way, could possibly reduce the consequences of chronic oxidative stress induced during vigorous running exercises in police dogs.

## 5. Conclusions

Feeding well-trained dogs exposed to rigorous exercise with fish oil-based food resulted in a significant improvement in total, LDL cholesterol, and glucose levels, which is consistent with literature data. Omega-3 LC PUFA is efficiently incorporated in plasma and erythrocyte phospholipids, resulting in improved cardiovascular health and immunity parameters in dogs. We proposed that changing regular dogs’ diets that are low in omega-3 PUFA to fish oil-based foods would significantly improve the oxidative status of dogs under chronically induced stress by decreasing the degree of lipid peroxidation and increasing GPx and CAT activity. This could, ultimately, positively promote dogs running exercises, as well as some other characteristics related to the purpose they have as active police dogs. The hypolipidemic and hypoglycemic effects of fish-based diets on dogs were as important as the positive results induced by alleviating oxidative stress after consuming fish-based diets.

## 6. Strengths and Limitations

The standard limitation of our study was the lack of a control group of dogs, which was explained by the fact that we had a limited number of police animals and that it was impossible to obtain some more.

## Figures and Tables

**Table 1 biomolecules-12-01092-t001:** The analytical composition of diets.

Fatty Acid and Macromolecules	Diet 1 (%)	Diet 2 (%)
PA, 16 0	23.14 ± 0.01	23.82 ± 0.70
PAO, 16 1	3.72 ± 0.26	4.76 ± 0.05 **
STE, 18 0	9.26 ± 0.26	6.63 ± 0.21 **
OA, 18 1 omega-9	40.61 ± 0.92	36.72 ± 0.83 *
VA, 18 1 omega-7	3.46 ± 0.33	2.98 ± 0.34
LA, 18 2	18.07 ± 0.26	15.44 ± 1.10
ALA, 18 3 omega-3	0.95 ± 0.02	2.41 ± 0.70 *
DGFLA,20 3	0.13 ± 0.02	0.10 ± 0.01
AA, 20 4	0.36 ± 0.12	0.44 ± 0.01
EPA, 20 5	0.04 ± 0.02	2.68 ± 0.09 ***
DTA, 22 4	0.13 ± 0.02	0.49 ± 00.04 ***
DPA, 22 5 omega-3	0.08 ± 0.01	0.35 ± 0.01 ***
DHA, 22 6	0.07 ± 0.02	3.17 ± 0.24 ***
SFA	32.40 ± 0.27	30.45 ± 0.86
MUFA	47.80 ± 1.33	44.46 ± 1.22
PUFA	19.80 ± 0.49	25.08 ± 2.21 *
Omega-6	18.69 ± 0.16	16.73 ± 0.06
Omega-3	1.14 ± 0.07	8.61 ± 1.04 ***
Omega-6/omega-3	16.39 ± 1.14	1.94 ± 0.05 ***
Proteins	25.74 ± 0.33	31.32 ± 2.80
Fat	16.23 ± 1.42	16.50 ± 0.35
Carbohydrates	46.03 ± 0.53	41.24 ± 4.79
Minerals	7.14 ± 0.74	8.31 ± 0.40

* *p* < 0.5, ** *p* < 0.01, *** *p* < 0.001, Student T test were used beetween diets.

**Table 2 biomolecules-12-01092-t002:** Changes in biochemical parameters after treatment.

Biochemical Parameters (mmol/L)	Baseline	After 3 Months
Glucose	4.82 (2.8–6.36)	3.26 (1.74–4.2) ***
Triglycerides	0.64 (0.31–1.03)	0.54 (0.46–0.94)
Cholesterol	5.86 (3.55–7.01)	4.47 (3.3–6.85) **
LDL	0.96 ± 0.62	0.54 ± 0.32 *
HDL	4.8 (3.32–6.79)	4.00 (3.16–6.03)

Values were presented as mean ± sd and as median with minimal and maximal values *** *p* < 0.001, *** p <* 0.01, * *p* < 0.05.

**Table 3 biomolecules-12-01092-t003:** Fatty acid profile in plasma phospholipids before and after treatment.

Fatty Acid (%)	Before	After
16 0	14.70 ± 1.06	17.63 ± 1.19 ***
16 1	0.51 ± 1.06	0.42 ± 0.07 ns
18 0	29.25 ± 1.81	29.00 ± 1.78 ns
18 1 omega-9	4.99 ± 0.48	6.03 ± 0.89 ***
18 1 omega-7	1.74 ± 0.22	2.25 ± 0.23 ***
18 2	24.80 ± 1.98	14.48 ± 3.47 ***
18 3 omega-6	0.40 ± 0.05	0.40 ± 0.04
18 3 omega-3	0.36 ± 0.09	0.27 ± 0.13 *
20 3	1.75 ± 0.46	1.12 ± 0.19 ***
20 4	18.45 ± 2.72	10.66 ± 1.85 ***
20 5	0.25 ± 0.09	7.42 ± 2.42 ***
22 4	0.79 ± 0.22	0.16 ± 0.04 ***
22 5	1.94 ± 0.45	2.19 ± 0.58
22 6	0.50 ± 0.22	7.97 ± 1.64 ***
SFA	43.66 ± 2.29	46.64 ± 1.49 ***
MUFA	7.24 ± 1.29	8.70 ± 1.15 ***
PUFA	48.84 ± 1.27	44.46 ± 1.41 ***
Omega-6	45.77 ± 1.14	26.82 ± 4.00 ***
Omega-3	3.04 ± 0.64	17.85 ± 4.00 ***
Omega-6/omega-3	15.60 ± 2.92	1.69 ± 0.90 ***
Omega 3 index	0.75 ± 0.26	15.39 ± 3.63 ***
20 4/20 5	82.58 ± 36.51	1.79 ± 1.22 ***

*** *p* < 0.001, * *p* < 0.05, ns-non significant.

**Table 4 biomolecules-12-01092-t004:** Fatty acid profiles in erytrocyte phospholipids before and after treatment.

Fatty Acid (%)	Before	After
16 0	17.14 ± 1.27	19.27 ± 0.79 ***
16 1	0.25 ± 0.11	0.36 ± 0.09 ns
18 0	29.84 ± 1.51	27.90 ± 1.03 ***
18 1 omega-9	7.12 ± 0.52	8.78 ± 0.92 ***
18 1 omega-7	1.88 ± 0.13	2.32 ± 0.24 ***
18 2	13.71 ± 1.03	8.55 ± 1.06 ***
18 3 omega-6	0.44 ± 0.08	0.38 ± 0.05 *
18 3 omega-3	0.22 ± 0.05	0.14 ± 0.04 ***
20 3	2.80 ± 0.33	2.08 ± 0.31 ***
20 4	23.35 ± 2.34	19.51 ± 1.88 ***
20 5	0.25 ± 0.06	6.04 ± 1.73 ***
22 4	1.58 ± 0.26	0.60 ± 0.29 ***
22 5	0.75 ± 0.18	0.91 ± 0.15 **
22 6	0.24 ± 0.15	3.24 ± 0.52 ***
SFA	46.98 ± 2.29	47.13 ± 0.97 ns
MUFA	9.49 ± 1.16	11.46 ± 1.01 ***
PUFA	43.34 ± 1.90	41.45 ± 1.41 **
Omega-6	41.89 ± 1.86	31.12 ± 2.98 ***
Omega-3	1.45 ± 0.29	10.33 ± 2.20 ***
Omega-6/omega-3	29.81 ± 5.15	3.24 ± 1.19 ***
Omega-3 index	0.49 ± 0.15	9.28 ± 2.07 ***
20 4/20 5	99.78 ± 26.63	3.65 ± 1.66 ***

*** *p* < 0.001, ** *p* < 0.01, * *p* < 0.05, ns-non significant.

**Table 5 biomolecules-12-01092-t005:** Changes in the content of EPA, DHA, and omega-3 PUFA in plasma and erythrocyte phospholipids before and after the treatment (Spearman correlation matrix).

	∆EPA e	∆DHA e	∆omega-3 PUFA e
∆EPA p	0.8382 ***	-	-
∆DHA p	-	0.5235 *	-
∆omega-3 PUFA p	-	-	0.6794 **

*** *p* < 0.001, ** *p* < 0.01, * *p* < 0.05, e: erythrocytes; p: plasma.

**Table 6 biomolecules-12-01092-t006:** Changes in oxidative stress parameters before and after the treatment.

	Before	After
TBARS (µmol/L)	26.5 ± 9.46	22.0 ± 7.15 *
	24.5 (8–49)	20 (12–46)
SOD (U/gHb)	2268.77 ± 1427.19	2577.72 ± 1861.45
	2374.2 (501–6253.5)	1925.1 (938–8635)
GPx (U/gHb)	299.68 ± 189.6	484.27 ± 233.5
	220.3 (94.4–596.6)	426.5 (246.4–1042.4) *
Cat (kU/gHb)	2.08 ± 0.70	2.55 ± 1.05 *

Values are presented as mean ± sd and as median with minimal and maximal values. * *p* < 0.05.

## Data Availability

The submitted manuscript to the journal is original has been written by the stated authors and has not be published elsewhere.

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
