# Peer review of "The Effect of Fish Oil-Based Foods on Lipid and Oxidative Status Parameters in Police Dogs"

_biomolecules, 2022, doi:10.3390/biom12081092_

Round 1

Reviewer 1 Report

The Authors revealed that the fish-based diet blood glucose, total, and LDL cholesterol levels in working (police) dogs were significantly reduced, indicating a positive biochemical profiles. In plasma and erythrocytes phospholipids omega-3 eicosapentaenoic and docosahexaenoic acid levels are increased and to contrary omega6PUFA as arachidonic acid are decreased causing a significant reduction in lipid peroxidation markers. 

This manuscript is more significant since it shows the importance of omega3PUFA intake as denoted in other studies. 

I have some questions:

1)   In the text, to insert the nomenclature of fatty acid (for example EPA, 20:5) because you use 20:5 in the table (the same DHA, ALA, DPA, AA and LA)

2)   In line 51, to delete the point after [6], the same in line 56 after [9] and in line 298 after [31]

3)   In line 81, to correct “comity” in “committee”

4)   In line 97, to delete the sentence: “The age of police dogs ranged from 3 to 7 years” because is written in line 88.

5)   The dogs were fed a commercial basic maintenance diet (Farmina dog foods) as indicated in line 102. Is the normal diet of dogs? I do not understand why you have fed the dogs for a month before the treatment with fish-based diet. The commercial basic maintenance diet is a deficient omega 3 PUFA and produces several metabolic alterations and maybe is not correct to do the comparison by utilizing it as a control.

In my opinion, it would be more correct to analysed also plasma and erythrocytes before the commercial basic maintenance diet.

6)   In table 1 you indicated the % of 16:0 in two diet and they are different, but in total SFA the % is similar, maybe is there a mistake? In general the principal SFA are palmitic acid and stearic acid and they are different

7)   In 157, to indicate the designation of “CP + A + CF + M” because only CP is reported in line 162 

8)   The 22:5 indicated in tables is omega3 or omega6?

9)   In line 279 , to indicate the designation of FO

10)                   From line 284 to 289, the Authors explained that omega-3 PUFA diet increased the omega-3 LC PUFA compensated by a significant decrease in omega-6 LC PUFA, but do not explain that there was a mechanism enzymatic competition between two families, as indicated in several studies.

11)                  In line 285 to correct “significant” 

12)                   In line 291, do not use “EM” or use together

13)                   in line 300, to use the “AA” in “arachidonic acid”

14)                   in line 331, to correct “20:4/20:5” with “AA/EPA” 

15)                  in line 375 to change “reactive oxygen species” with “ROS”

16)                  in line 384, to correct “OMEGA” with “omega”

Author Response

Dear Reviwer 1,

thank you for your suggestions and your opinion about our study and presented results. We tried to answer the things that you addressed and hope tha tyou are satisfied with it.

Sincerely

dr Tamara Popovic

Reviewer 2 Report

160722

I have reviewed the manuscript by Ravic and colleagues entitled “ The effect of fish-based foods on lipid and oxidative status parameters in police dogs“. This is an interesting study which documents the modulation of lipid homeostasis and oxidative status in the circulation of police dogs following dietary supplementation with EPA and DHA by means of a 3 month feeding with a fish oil-enriched food. Whereas the manuscript is generally well presented it lacks precision. It will also benefit from a general English grammar revision. I provide the following comments which the authors may want to address to improve their manuscript.

Comments:

Title: “fish-based food” should be changed to “fish oil-based food”. According to the description of the supplemented dog food, the food was enriched with fish oil and not with fish.

Line 50 – please refer to “..with marine fish rich in DHA”. Not all marine fish species are rich in DHA, and not all DHA-rich fish species are so throughout the year.

Line 68 – thromboxane A2 is an endoperoxide but it is not a lipid peroxidation product. Why do you mention it here in the context of oxidation and oxidation products?

Line 71 – you suggest that omega-3 fatty acids make membranes more permeable to nutrients, suggesting to allowing enhanced diffusion across membranes. But please use the concept of transmembrane transporters for nutrients as most nutrients are not distributed by diffusion across membranes.

Line 101 and 104 – Please provide the specific product name of this Farmina dogfood, and for the fish oil-enriched Farmina dog food, as well as the city and country where the Farmina manufacturer is located. If it is not a commercial product, please provide a detailed description how the food was prepared. Think about which information should you provide so that another research group could replicate your work. It would be interesting to indicate the specific fish oil (type, producer) that was used, as well.

Line 106 – Do you know, or have you measured the tocopherol content of the two types of dogfoods used?

Table 1:

-       Please calculate the statistical significance of any differences

-       The parameter % should be mentioned on top of column 2 and 3, not column 1 which is the name of the fatty acid

-       Column 1 cannot be named “Fatty Acid” as it also reports on other parameters. Perhaps the authors can use subheaders or divide the table in portions.

-       The number of replicates used to calculate the mean and standard deviation should be reported. In the methods section it should be explained which method was used for the fatty acid composition, the number of replicates that are reported in this table, and in which laboratory the measurements were carried out.

-       There is a huge increase in the relative proportion of palmitic acid in Diet 2. Has the value been correctly entered in table? Or is it wrongly put in Diet 1? Please double check.

-       The total fatty acid percentage in column 2 (Diet 1) amounts to 80.02%. For diet 2 it is 99.99%. Why is it not (close to) 100% for diet 1?

Line 154 – Please provide a method reference that clearly describes the method performance and execution. Otherwise give a detailed description of the GC temperature profile, the use of split or splitless injection, the liner type, injection volume, carrier gas, etc.

Line 155 – I think it should be made clear that calculations were based on detector area percentages, not weight percentages.

Line 157 – Please define A, CF and M (CP is defined a little below)

Line 168-175 – This section needs a method reference. Or a detailed description of all volumes used and a definition of “amount of enzyme”

Line 179-185 – A method reference is needed, and preferably a better description of the method principle – it is not clear how GPX is measured if GSH reductase is apparently competing for NADPH

Line 191 – please change “..all the enzymes” to "..catalase, GSH peroxidase and SOD”

Line 209 – Please rephrase this sentence to make it clear for which purpose this correlation analysis was employed in your study.

Line 221 – This phrase does not belong in the Results section. Please move to the Discussion.

Table 2 – You can consider renaming the column titles from “Before” and “After” to “Baseline” and “3 months”

Table 2 – Typo – rewrite to triglycerides,

Line 227 – 230 – These are not results, please move to the Discussion section.

Line 332 – The results do not say anything about quantity, only about the profile (relative proportions) of specific fatty acids in plasma or in RBC membrane phospholipids. Please correct.

Line 332 – please insert “(Table 3)” after “increased in plasma”

Line 332 – please insert “(Table 4)” after “in erythrocyte membrane”

Line 234 -235 – the section of the phrase “…indicating …to the end of sentence,  are not results but an interpretation. Please move to the Discussion section, and discuss there.

Line 270 – change positive to negative regulation, or change insulin resistance to insulin sensitivity

Line 279 – According to your methods description (line 104), the Farmina diet type 2 is enriched in fish oil, not in fish. So please correct “Our fish-based food..” to “Our fish oil-supplemented food…” . Same for line 342, line 364 and line 382 and 387. It is very critically important that you describe properly in the Method section if your Type 2 diet is supplemented with fish meat or fish oil.

Line 282 – You could also mention the triglyceride-lowering effect of supplemental EPA/DHA in human volunteers, especially in those with hypertriglyceridemia.

Line 283 – Discuss this also in light of the diet composition. Only palmitic acid is significantly higher but overall MUFA are not. Is palmitic acid increased in plasma and RBCs? Discuss how this is possible.

Line 288 – It is not clear what is meant by “agent-inducing changes”. Please clarify.

Line 294 – you mention the high EPA content, but content of specific fatty acids was not measured in your study --- What was measured is the fatty acid profile, which are relative proportions of fatty acids, but says nothing about their content or levels.

Line 304 – Please provide a correct definition of the Omega-3 Index, i.e. sum of EPA and DHA as percentage of the total fatty acid composition of RBC membranes

Line 332 – the generalized statement regarding can only be made if you provide one or more references, since your study did not address inflammation.

Line 337 -338 – Your study did not address lipid peroxidation in liver microsomal lipids. Please remove this supposed “concordance” and limit the discussion and conclusions to what you have studied.

Line 339-343 – The wording “clearly showed” is an overstatement, since only MDA was measured as a readout of lipid peroxidation. Many other aspects of lipid peroxidation could have been measured, but were not addressed. Your results are still far from clear, and you could instead be modest about the preliminary but very relevant evidence you have collected.

Line 343 – please replace “lipid peroxides” with “of superoxide anion radical, hydrogen peroxides, and lipid hydroperoxides”

Line 343-349 needs a reference that provides insight into that mechanistic explanation

Line 348 – less susceptible than what alternative configuration? Nearly all of EPA and DHA in biological systems is esterified, mostly within phospholipids. Which other chemical forms are less stable? Please be precise and use references that support your discussion.

Line 350 – you can remove the word “respectfully”.

Line 352 – If I understand it well, these dogs were already receiving the same level of training before the feeding period started. The explanation that MDA would suddenly decrease more than before is not likely. Please rewrite.

Line 356 – I don’t agree that finding that TBARS decrease after supplementation is a limitation of your study. It is one of the main results. If you want to dedicate some lines to the limitations of your study, focus on the lack of a control group, but defend your results. You could also discuss what new studies you plan to do to support the finding that supplemental EPA/DHA intake might reduce systemic lipid peroxidation

Line 363 – Not “invariably”- the level of membrane perturbation depends on the level of peroxidation vs the antioxidative capacity. If it would be invariable, all living systems would be permanently in a state of lipid peroxidation (not compatible with life)

Author Response

Dear review 2 ,

thank you very much for your kind suggestions and a lot of very usuful comments. We tried to fulfil and do everything you asked in best possible way.

We hope it will implove our article to be much better and more understandable for reading.

Sincerely

Dr Tamara Popovic

Round 2

Reviewer 1 Report

in my opinion the manuscript would be perfect, but only two minor mistakes:

-in table 1 to write the fatty acid as in the other tables

-in line 264 the Authors reported fish-oil based food (FO), in my opinion would use together  FO or delete (FO).

Author Response

Dear Reviewer,

Thank you for all suggestions and opinion. We did all you asked and I just emphasised one fact in answers which you will I hope have in mind about Table

We are thankful for our collaboration and wish all the best to you

Dr Tamara Popovic